

# Concept for an electrostatic focusing device for continuous ambient pressure aerosol concentration

Joseph L. Woo[1,2*], Neha Sareen[1,3], Allison N. Schwier[1,4], V. Faye McNeill[1*]

[1]Department of Chemical Engineering, Columbia University, New York, NY, 10027, USA
5  [2]Current affiliation: Department of Chemical and Biomolecular Engineering, Lafayette College, Easton, PA, 18042, USA
[3]Current affiliation: US Environmental Protection Agency, New York, NY, 10007, USA
[4]Current affiliation: US Department of State, Washington, DC, 20520, USA

*Correspondence to*: Joseph L. Woo (wooj@lafayette.edu) and V. Faye McNeill (vfm2103@columbia.edu)

10  **Abstract.** We present a concept for enhancing the concentration of charged submicron aerosol particles in a continuous flow stream using *in situ* electrostatic focusing. It is proposed that electrostatic focusing can enable the continuous, isothermal concentration of aerosol particles at ambient pressure, without altering their chemical composition. We model this approach theoretically and demonstrate proof-of-concept via laboratory measurements using a prototype. The prototype design consists of a nozzle-probe flow system analogous to a virtual impactor. The device was tested in the laboratory using submicron, 15  monodisperse stearic acid particles. Particles were charged using a unipolar charger, then concentrated using a cylindrical electrostatic immersion lens to direct the charged submicron particles into the sample probe. Under applied lens voltages ranging from 0V to 30kV, aerosol concentration increased up to 15%. Observed particle enrichment varied as a function of voltage and particle diameter. These results suggest that an imposed electric field can be used to increase aerosol concentration in a continuous flow. This approach shows promise in increasing the effective enriched size range of virtual impactors or other 20  continuous-flow methods of collection.

## 1 Introduction

It is desirable for a number of applications to be able to continuously concentrate submicron aerosols at atmospheric pressure without affecting their chemical composition. For example, the leading techniques for detailed chemical characterization of the organic fraction of ambient particulate matter rely on particle collection-volatilization cycles (Williams et al., 2006; Lopez-25  Hilfiker 2014). Online (continuous flow) aerosol concentration would improve instrument time response and simplify data analysis.

Online aerosol concentration techniques typically involve inertial concentration. The aerodynamic lens approach (Liu et al., 1995a; Liu et al., 1995b) has been effective for aerosol mass spectrometry applications (Jayne et al., 2000), but the low ultimate pressure necessary for collection of 50-500 nm particles (Wang and McMurry, 2006) makes an aerodynamic lens inlet 30  unsuitable for use with some detection techniques such as CIMS (Hearn and Smith, 2004; Sareen et al. 2010, Lopez-Hilfiker



et al. 2014). The virtual impactor approach (Chen et al., 1986) is effective for larger particles, but less successful for concentrating submicron aerosols due to their low Stokes numbers (Seinfeld and Pandis, 2016). This has been addressed in the past via condensational growth of the particles prior to concentration (Kim et al., 2001a,b, Gupta et al., 2004, Geller et al. 2005), which can alter the measured aerosol organic composition (Su et al., 2006; Jung et al. 2010). Another possible approach

is to use high aerosol flow velocities. High flow velocities can lead to nonisothermal conditions, which are undesirable for analysis of semivolatile aerosols. Near-isothermal flow may obtained by ensuring that the flow does not decelerate during sampling (Dhaniyala et al., 2003) or by restricting flow velocities to below the compressibility limit. Lee and coworkers (2003) presented and characterized a nanoparticle virtual impactor which used a convergent-divergent (de Laval) nozzle to achieve high flow velocities without shockwaves (Lee et al., 2003), achieving a 50% cutoff diameter of approximately 60 nm when

operating at ambient pressure with an impactor chamber maintained at 220 Torr. Thermal performance was not characterized in that study. Middha and Wexler (2006) built upon the Lee et al. inlet by designing a virtual impactor with a slot-shaped nozzle, which is the conceptual equivalent of several cylindrical nozzles functioning in parallel. This design was intended to produce high throughput and allow higher operating pressures than a single axisymmetric nozzle. Flow was nonisothermal in that device, with a predicted temperature gradient of ~20K.

In this study, we present a concept for the application of *in situ* electrostatic focusing to isothermally concentrate a continuously flowing aerosol stream of submicron particles, at ambient pressure, without altering their chemical composition. Electrostatic precipitation has been used for decades to remove charged particles from gas streams and enhance particle collection for offline measurement techniques (Liu et al., 1967, Dixkens and Fissan, 1999, Fierz et al. 2007). We propose an approach based on a virtual impactor geometry (Chen et al. 1986), such that an enriched aerosol stream can be continuously generated, e.g. for

online characterization. A two-component cylindrical immersion lens is placed concentric to the gas-particle separation region (Figure 1). The particles are charged upstream of this region, and then passed through a radially symmetric, inward-focusing electric field in order to focus them into a sampling probe. We model this approach theoretically and demonstrate proof-of-concept via laboratory measurements using a prototype.

**2 Design Considerations**

As described in the previous section, in order to create an aerosol concentrator suitable for use upstream of Aerosol-CIMS or similar techniques, the following design requirements must be met: 1) concentration of submicron aerosol, 2) ultimate pressure of ≥ 2 Torr, 3) isothermal or near-isothermal sampling, and 4) no chemical alteration of particles from the sampling process. Our approach is based on the concept that aerosol particles in a virtual impactor may deviate from the surrounding gas flow, depending on the balance of inertial, electrical, and viscous drag forces acting on the particle. For a charged particle in an

electric field, the particle equation of motion is expressed as follows (Seinfeld and Pandis, 2016):

$$m_p \frac{dv}{dt} = \frac{3\pi\mu D_p}{C_c}(u - v) + qE \qquad (1)$$



where $m_p$ is the particle mass, $\boldsymbol{v}$ is the velocity vector of the particle, $\mu$ is the gas viscosity, $D_p$ is particle diameter, $C_c$ is the slip correction factor, $\boldsymbol{u}$ is the velocity vector of the carrier gas, $q$ is the charge on the particle, and $\boldsymbol{E}$ is the electric field vector. After non-dimensionalizing the above equation, we can derive a dimensionless parameter, $\kappa$, which describes the balances between electrostatic (deflective) and inertial forces. The derivation of $\kappa$ can be found in the Supporting Information.

$$\kappa = \frac{Z \cdot E_0}{St \cdot u_0} = \frac{6LqE_0}{\rho \pi D_p^3 u_0^2} \tag{2}$$

Here, $L$ is a characteristic length of the aerosol flow stream, set in this study as the gap distance between the orifice and collection probe defining our virtual impactor flow system. $Z$ is the electric mobility of the particle, and $\rho$ is the density of the aerosol particle. $E_0$ is a characteristic electric field strength. When $\kappa$ approaches or exceeds 1, electrostatic forces become sufficient to deflect submicron particles toward the sample probe. Note that for $\kappa \gg 1$ over-deflection and particle losses may occur. In Figure 2, $\kappa$ is plotted as a function of electric field strength for a 30 nm particle flowing at a range of subsonic velocities. A variety of combinations of the variables satisfy $\kappa \geq 1$.

**2.1 Flow Characterization**

Performance of the concentrator prototype was simulated in more detail using a numerical model that approximates the force balance on a particle in the proposed prototype geometry (see the following section for details). The trajectory of single particles through the concentrator was modeled, assuming gas flow characteristics of an inertial impactor with cylindrical geometry. Flow was approximated using potential flow of an incompressible gas into a stagnation point with dimensionless radial and axial coordinates, $\hat{r}$ and $\hat{z}$ (Bird et al. 2007):

$$\widehat{u_r} = \hat{r} \tag{3}$$

$$\widehat{u_z} = \frac{1}{2}(1 - \hat{z}) \tag{4}$$

Here, $\hat{r}$ and $\hat{z}$ are dimensionless radial and axial positions scaled to the lens radius and half of the gap distance between the entrance orifice and collection probe, respectively. The origin of $\hat{r}$ and $\hat{z}$ was set to be the midpoint between the two cylindrical lenses, at the center of the axis of symmetry of the lenses (see Figure 1). $\widehat{u_r}$ and $\widehat{u_z}$ are the dimensionless radial and axial velocities, both scaled to the initial entrance velocity. The initial trajectory of the gas-aerosol stream was defined within an orifice entrance region, described in further detail below, with an intial velocity of $(\widehat{u_r}, \widehat{u_z}) = (0,1)$.

**2.2 Electrical Deflection**

We use a cylindrical immersion lens as a source of generating an electric field, though any geometry that is capable of generating radially inward electric fields while not interfering with bulk flow may be used. Immersion lenses are a commonly used method of manipulating streams of low-energy electrons and particle beams (Páris 1966; Heddle 2000; Humphries 1999; Read 1971; Gillespie and Brown 1997). By placing two charged conductive cylinders of equal radius in close proximity to one



another, a radially symmetrical field is generated as a function of its geometry and the applied voltage gap between the two lenses.

A charged particle moving in the axial direction of the cylinders that enters this field is subject to two deflective forces: an axial accelerating force, and a radial force that pushes both towards and away from the axis of symmetry. The net effect of these forces is a deflection of the overall path of the particle towards the center line and the entrance of the collection probe. A cylindrical lens assembly can be positioned over a virtual impactor geometry such that the submicron particles exiting the nozzle can be electrically deflected toward the collector (minor flow), without otherwise interfering with the gas flow field. The gap distance between the two component lenses was set to match that of the previously described virtual impactor gap distance.

The electric field generated from this assembly can be described as the gradient of the applied voltage field resulting from the immersion lens, which was approximated by Bertram (1942) for our geometry as:

$$\hat{V} = 0.15\big(X_0(\hat{r}, \hat{z}+1) + X_0(\hat{r}, \hat{z}-1)\big) + 0.7\big(X_1(\hat{r}, \hat{z}+1) + X_1(\hat{r}, \hat{z}-1)\big) \qquad (5)$$

$X_0$ and $X_1$, seen in Table 1, are functions of the zeroth and first Bessel functions of the first kind, $J_0$ and $J_1$. In these equations, $\mu$ represents the set of positive roots of the zeroth order Bessel function, such that $J_0(\mu) = 0$.

One thousand particle trajectories were modeled with and without the presence of applied voltage. The trajectories were initated at the outlet of the orifice ($\hat{z} = -1$), with radial positions spaced equally from $0 < \hat{r} < 0.1$ (see Figure 1). A particle was considered to have been captured by the collection probe if its trajectory terminates within the collection region ($\hat{z} = 1, \hat{r} < 0.22$).

The ratio of captured streams with and without applied electric fields, referred to as an *enrichment ratio*, was used to quantify the predicted performance of the prototype for a given set of conditions. Figure 3 shows the calculated enrichment factors for singly charged particles of typical aerosol density (1 g cm$^{-3}$).

**3 Prototype design**

A schematic of the relevant geometry of the prototype is shown in Figure 1, with key dimensions listed in

Table 2. Two lengths of 0.444 cm inner diameter stainless steel tubing were attached into opposite ends of a KF-16 five-way cross vacuum fitting to act as an aerosol flow inlet and collection probe, respectively. A 0.1 cm inner diameter tube was bonded to the end of the inlet tube, forming a flow restriction. The ends of the inlet and collection probe were set 0.1 cm apart from one another, forming a gap for non-captured flow to travel through. The two junctions coplanar to the inlet and collection probe were connected to an exhaust, providing an exit for the major flow. All of these components were electrically connected to one another and grounded.



The lens assembly consists of two electrically isolated, 1.0 cm inner diameter, stainless steel tubes mounted with their ends in close proximity to one another, matching the gap distance between the two flow tubes. A gas-tight plug fitted with two electrically isolated electrodes was installed on the top junction of the KF-16 five-way cross, and wired to each of the two tubes. The lens assembly and the two flow probe tubes were wrapped in PTFE tape to eliminate potential electric field

interactions in other parts of the concentrator setup. Voltage was applied to the two cylindrical lenses using a high voltage power source (Kepco). Under conditions without applied voltage, lenses were electrically grounded.

## 4 Performance Assessment

The electrostatic focuser was tested using a monodisperse stream of stearic acid particles produced via homogeneous nucleation. The experimental setup for testing is shown in Figure 4. Ultra-high purity nitrogen (TechAir) was introduced into

a ¼" ID glass tube filled with solid stearic acid (Fisher Scientific). This tube was wrapped with a heating tape and brought to 120°C using a variac (Staco Energy). The generated particles were then passed to a tube with a second heated region maintained at 120°C. Moving the location of the secondary heated region allowed for fine control of the peak diameter of the generated stearic acid aerosol. This aerosol stream was maintained at 1.5LPM, and size-selected using a differential mobility analyzer (DMA, TSI Model 3080).

The monodisperse aerosol was then passed through a unipolar corona charger, producing a stream of positively-charged particles. This stream was then diluted to a total flow of 15 LPM and introduced to the electrostatic focuser. The outflow of the collection probe tube was maintained at 1.5LPM, with excess flow travelling to exhaust.

Particle concentrations were traced at 1-second resolution using a condensation particle counter (CPC, TSI Model 3775), with and without applied voltage. (An example of such a trace can be seen in Figure 5). Enrichment factors were evaluated by

comparing the time-averaged total aerosol concentrations of the collection probe outflow. At each condition, particle size distributions were also checked for consistency, using a scanning mobility particle sizer (SMPS, GRIMM Inc).

The effect of the prototype concentrator on Aerosol-CIMS signal was also tested, using a custom-built mass spectrometer. Further description of this setup can be found in Sareen et al. (2010). After passing through the electrostatic concentrator, stearic acid aerosol was passed through a 23 cm-long, 1.25 cm ID stainless steel tube wrapped in heating tape, to volatilize

aerosol input before entering the chemical ionization region. The external temperature of this volatilization region was maintained at 150 °C using a thermocouple and temperature controller (Staco Energy). Gas-phase analyte molecules were detected as products of their interactions with $I^-$ reagent ions (m/z values of 284 ($C_{18}H_{35}O_2^-$), 319 ($C_{18}H_{35}O_2^- \cdot 2H_2O$), and 409 ($I^- \cdot C_{18}H_{36}O_2$)) using a quadropole mass spectrometer (Extrel).

Both the SMPS and Aerosol-CIMS used in this study sample at 1.5LPM. All other flows of the electrostatic focuser were

maintained using needle valves and monitored using a mass flow meter (MKS Instruments). The major (exhaust) flow was controlled using a throttled mechanical pump (Varian DS302) and monitored with a separate, volumetric gas flow meter.



## 5 Results and Discussion

When voltage is applied to the outlet lens (vis. Figure 1), an increase in the particle concentration of the minor flow stream is observed (Figure 5). The observed enrichment is summarized in Figure  as a function of applied voltage. As shown in Figure , at 30 kV applied voltage the dependence of the observed enrichment on particle size is opposite that predicted due to the

model.  This may be a result of over-deflection of the particles with the highest mobilities (i.e., the smallest particles) via the imposed electric field. No discernible change in sampled particle concentration was observed when voltage was applied to the inlet lens. Subsequent references to applied voltage in this discussion will refer to that applied the outlet lens only.

We note that, while its design was based on virtual impactor geometry, at the pressure and flow conditions used here, the concentrator prototype does not operate as a true virtual impactor for submicron particle sizes. That is, with no applied voltage

there is no inertial concentration of submicron particles and the particle concentrations in the inlet, major and minor streams were identical within the noise. Hence, the results shown here successfully demonstrate the isolated effect of electrostatic aerosol concentration.

Aerosol-CIMS traces in the absence and presence of voltage indicated enrichment consistent with the mass enrichment that was observed via the SMPS. No additional signal peaks beyond those of stearic acid were observed in Aerosol-CIMS during

operation regardless of applied voltage, implying that stearic acid did not undergo any chemical reactions as a result of the charging process or from exposure to the electric field of the electrostatic focuser. Corona discharge-based unipolar chargers, such as the one employed in this study, are known to generate ozone at concentrations as high as 5ppm near the filament (Wang and McMurry 2006). Unlike stearic acid, unsaturated organic species such as oleic acid could therefore be vulnerable to $O_3$ oxidation within the device. The expected residence time inside the unipolar charger used in this study is estimated at less than

10s, and the charger outflow is diluted by a factor of 10, reducing the potential impact of oxidation.  However, the risk of unwanted chemical oxidation could be eliminated by using an alternative method of imposing charges onto the input aerosol stream that does not generate ozone (e.g. Han et al., 2008).

Calculated values for the Mach number reach as high as 0.94 within the entrance orifice, beyond the generally accepted threshold of incompressible flow (Kundu, 1990). However, due to the relatively short length scale of the entrance orifice, the

predicted ratio of pressures across the inlet entrance is expected to be sufficiently low (~1.05) to avoid choked flow (Miller 1996). Furthermore, temperature at various locations within the prototype was measured using a thermocouple after extended operation, and no significant temperature fluctuations were observed. Finally, Aerosol-CIMS traces for stearic acid in the absence of volatilization did not show any deviation from background values as a function of the flow velocity through the concentrator, consistent with this conclusion.

At all voltages tested, observed enrichment was lower than model predicted values under the same conditions. The aerosol enrichment observed via SMPS and Aerosol-CIMS corresponded to that predicted for voltages that were between 15-20% of those actually applied.  We believe that this discrepancy between model and observation is largely due to distortion of the





electric fields within the gas-particle separation region. Additional minor losses could be due to particle losses inside the sample probe.

The likely sources of electric field distortion are solid surfaces within the lens assembly, especially the collection probe, due to its close proximity to the outlet lens. As previously mentioned, the collection probe was covered with PTFE, but this likely

provided incomplete electrical insulation. In order to evaluate the potential impact of field distortion on the performance of the electrostatic focuser, we estimated the magnitude of the electric field generated by the potential between the outlet lens and the collection probe in a worst-case scenario in the absence of any insulation (see the Appendix for details of the calculation). We approximated the fringe effects of the two concentric cylinders (the outlet lens and the collection probe) as being similar to that derived from two parallel plates. Several studies have examined the effective field strength, voltage distribution, and

charge densities of the edges of finite plates of infinite width (Pillai 1970; Parker 2002). The two relevant voltage distributions, described by Bertram (1942) (inlet lens-outlet lens) and Parker (2002) (outlet lens-collection probe), were combined in order to estimate their relative effects on the total electric field. In the wost-case scenario of an uninsulated collection probe, the magnitude of the radial component of the combined electric field becomes negligible within the outer radius of the collection probe. That is, electrostatic enrichment of the aerosol becomes negligible. These calculations assume that the collection probe

is exposed and conductive. Taking measures to decrease its behavior as a source of undesired electric fields, such as the PTFE tape insulation employed in this study, will help control distortion and lead to enrichments closer to theoretical values.

Fierz et al. (2007) used electrostatic focusing to enhance particle collection via inertial impaction onto a TEM grid in a radially symmetric design reminiscent of the one employed here. They achieved 7-9% collection efficiency for 50nm aerosol at an applied voltage of 3kV, which is close to our theoretical enrichment (12%) at the same voltage. We expect that the electric

field in their system, which generated via the potential between a lens and their collection plate, was relatively free of distortion, due to the absence of additional nearby solid surfaces in the gas-particle separation region (which would not be possible in a continuous flow design such ours). This provides further support for the notion that distortion of the electric field was the primary cause of the observed discrepancy between the model predictions and observed enrichment.

It should be restated that the cylindrical lens geometry in the design used here was implemented because of its relatively simple

construction and lack of interference with the gas flow field. For the desired focusing effect, a net radially-inward component of the electric field is needed, but the exact geometry of the electric fields, and the elements needed to generate them, are non-specific. As our experimental results have indicated, so long as a sufficiently strong electrical deflective force can be placed opposite to an inertial flow force in the radial direction, modulation and concentration of an aerosol stream can be achieved.

Pressures inside the aerosol concentrator were kept at ambient or positive values in this study, but our model outputs suggest

that this geometry can be implemented at lower pressures without affecting the net deflection forces of an imposed electric field. However, lower gas pressures inside the charging region can decrease the effective breakdown voltage between the two cylindrical lenses of this system, increasing the probability of arcing.



## 6 Summary and Conclusions

We have demonstrated a novel concept for enhancing the concentration of submicron charged aerosol particles in a gas stream by means of electrostatic focusing. Stearic acid aerosol concentration enrichments up to 15% were observed using a proof-of-concept prototype under applied voltages up to 30kV. This approach is compatible with analytical methods that require

isothermal, ambient pressure sampling and relatively high ultimate pressures. Our prototype employed a cylindrical immersion lens and a virtual impactor geometry. Although flow velocities used here were too low for concentration of submicron aerosol aerodynamically using virtual impaction, electrostatic focusing could enhance the performance of a virtual impactor, allowing the concentration of smaller particles. We note that, besides a cylindrical immersion lens, electrostatic focusing of particles may be also achieved by a charged mesh tube, cylindrical lens assembly (e.g. Einzel lens), octupole, or other means of

generating a radially inward-facing electric field.

### Acknowledgements

The authors acknowledge the NSF BRIGE program (EEC-0823847) for funding.

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

.


**Table 1 - Cylindrical Lens Functions X0, X1 (adapted from Bertram, 1942).**

| | $1 > \hat{z} > 0$ | $-1 < \hat{z} < 0$ |
|---|---|---|
| $X_0(\hat{r}, \hat{z})$ (Eqn. 4a, 4b) | $1 - \sum_{+\mu} \dfrac{J_0(\mu\hat{r})}{\mu J_1(\mu)} e^{-\mu\hat{z}}$ | $\sum_{+\mu} \dfrac{J_0(\mu\hat{r})}{\mu J_1(\mu)} e^{\mu\hat{z}}$ |
| $X_1(\hat{r}, \hat{z})$ (Eqn. 4c, 4d) | $\hat{z} + \sum_{+\mu} \dfrac{J_0(\mu\hat{r})}{\mu^2 J_1(\mu)} e^{-\mu\hat{z}}$ | $\sum_{+\mu} \dfrac{J_0(\mu\hat{r})}{\mu^2 J_1(\mu)} e^{\mu\hat{z}}$ |

**Table 2 - Prototype Concentrator Dimensions**

| Component | Length (cm) |
|---|---|
| **Inlet Probe Orifice Diameter** | 0.1 |
| **Inlet Probe Outer Diameter** | 0.635 |
| **Inlet Probe Orifice Length** | 0.145 |
| **Orifice-Collection Probe Gap Distance** | 0.1 |
| **Collection Probe Inner Diameter** | 0.444 |
| **Collection Probe Outer Diameter** | 0.635 |



| | |
|---|---|
| **Immersion Lens Inner Diameter** | 1.0 |
| **Immersion Lens Outer Diameter** | 1.28 |
| **Orifice Approach angle** | 90° |

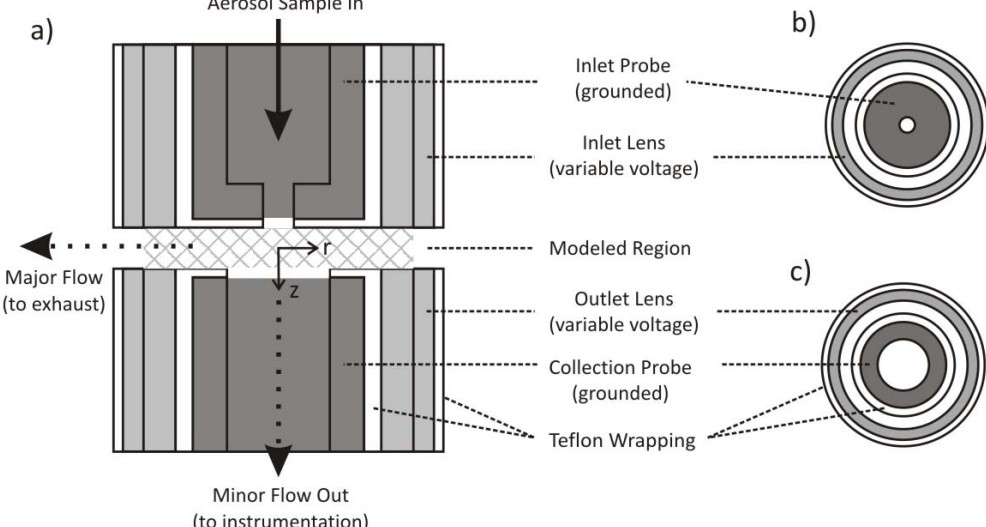

5    **Figure 1: Schematic of the prototype electrostatic concentrator perpendicular to flow (a), down the inlet probe (b), and down the collection probe (c). Dark grey regions are grounded, white regions are PTFE insulation, and light grey regions are at variable voltage.**





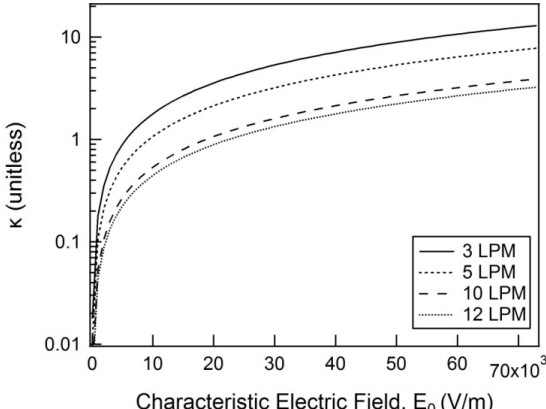

**Figure 2: κ vs. Voltage (V) for a range of velocities that satisfy the AC-CIMS design requirements, using prototype geometry. Particle diameter 30nm.**

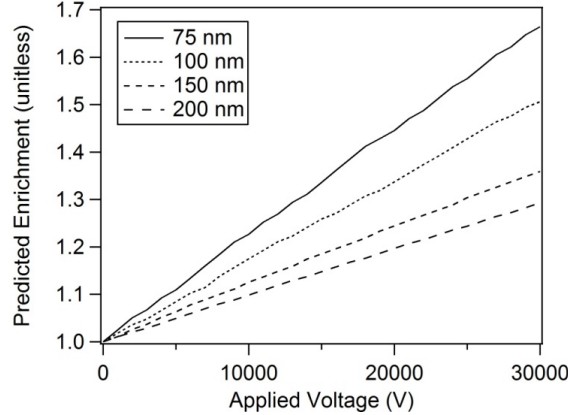

**Figure 3: Predicted Enrichment versus applied voltage under experimental operating conditions.**





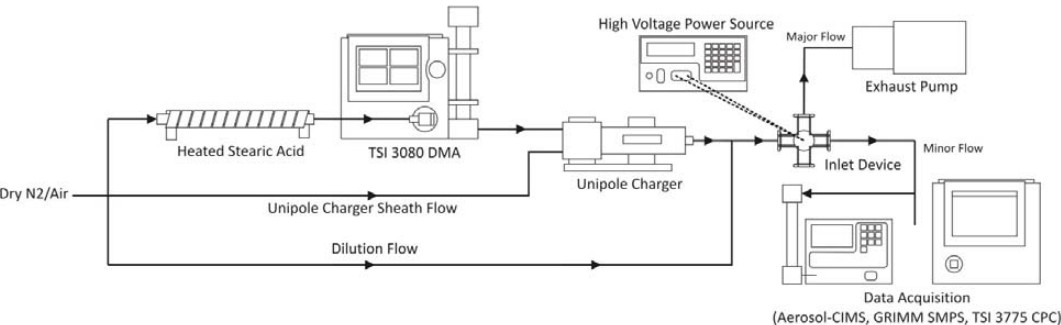

**Figure 4: Experimental setup for evaluation of concentration gain from the electrostatic focuser.**

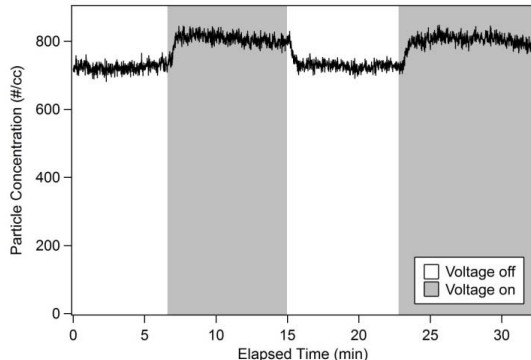

5    **Figure 5: SMPS particle concentration trace with and without applied voltages. 200nm, 30kV applied voltage.**





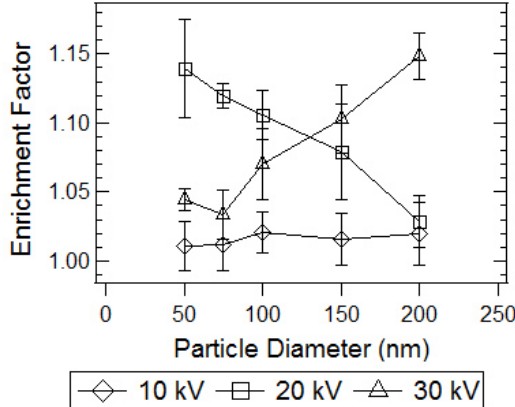

**Figure 6: Observed aerosol enrichment at varying voltages and particle diameters.**

