# Peer review of "Concept for an electrostatic focusing device for continuous ambient pressure aerosol concentration"

_Atmospheric Measurement Techniques, 2019_

## Referee Comment (RC1) · Anonymous Referee #2 · 18 Mar 2019

The paper presents an approach to enhance the aerosol concentration in a continuous flow, similar to virtual impactors that are commonly used in the aerosol science field. Theoretical calculations are performed. A concept is designed, and a prototype based on this design is developed and tested. Size-selected stearic acid particles are used to test the performance, and it was shown that up to 15% enhancement can be achieved.

The paper fits within the scope of AMT. It provides a concept of using electric fields to enhance the concentration, and results from one test experiment is presented to show that the concept is working. My primary concern is that these only one test experimental data is not sufficient to conclude that concept can actually be used for general

aerosol studies where enhancement is necessary for aerosol detection. One cannot reach the conclusions that enhancement is possible with just one test experiment. I would like to refer to the paper in AMTD by Saarikoski et al. (2019), which focuses on the exact problem the current paper is focused, although the technique is different. The presented concept needs to be tested using different aerosol systems, and validation should be confirmed from understanding size-distribution and chemical composition of aerosol (minor flow output). Such aerosol properties should be reported. Also, note that Saarikoski et al. (2019) paper shows the concept and numerous lab and field data to show that submicron aerosol enhancement is possible without electrostatic focusing. For all these reasons the paper is not ready for the publication and rated poor in terms of scientific significance. I recommend adding more results and resubmit.

Saarikoski, S., Williams, L. R., Spielman, S. R., Lewis, G. S., Eiguren-Fernandez, A., Aurela, M., Hering, S. V., Teinilä, K., Croteau, P., Jayne, J. T., Hohaus, T., Worsnop, D. R., and Timonen, H.: Laboratory and field evaluation of the Aerosol Dynamics Inc. concentrator (ADIc) for aerosol mass spectrometry, Atmos. Meas. Tech. Discuss., https://doi.org/10.5194/amt-2019-74, in review, 2019.

---

## Author Comment (AC1) · 3 Apr 2019

We thank the reviewer for their comments. We also thank them for bringing the recently published Saarikoski et al. (2019) to our attention. We will be sure to cite that work in the revised manuscript.

*My primary concern is that these only one test experimental data is not sufficient to conclude that concept can actually be used for general aerosol studies where enhancement is necessary for aerosol detection. One cannot reach the conclusions that enhancement is possible with just one test experiment.*

[Figure]

Perhaps this was not sufficiently clear from the Discussion Paper, but the apparatus was tested in multiple test experiments for a range of operating conditions. Test experiments were performed to quantify aerosol concentration enrichment for a range of applied voltages and particle sizes. The observed enrichment achieved by our electrostatic focuser, encompassing multiple replicates of a total of fifteen operating conditions, is summarized in Figure 6. We will edit the manuscript to ensure that this point is clear in the final version.

*The presented concept needs to be tested using different aerosol systems, and validation should be confirmed from understanding size-distribution and chemical composition of aerosol (minor flow output).*

Monodisperse aerosol at different selected particles sizes (via size selection through a DMA, see Figure 6) was used to demonstrate effective concentration as a function of particle size, in more a precise manner than would have been achieved using polydisperse aerosol. Our focus is on organic aerosol, as discussed in the introduction, and homogeneously nucleated stearic acid was chosen as a representative aerosol type. We don't expect electrostatic focusing of other organic aerosol types to vary significantly from stearic acid aerosol. Chemical composition of the minor flow output, as measured via Aerosol-CIMS, was pure stearic acid, as expected (cf. page 5, lines 26-28 and page 6, lines 14-16 "No additional signal peaks beyond those of stearic acid were observed in Aerosol-CIMS during operation regardless of applied voltage, implying that stearic acid did not undergo any chemical reactions as a result of the charging process or from exposure to the electric field of the electrostatic focuser").

In response to the reviewer comment we will revise the manuscript to further emphasize our focus on organic aerosol in the introduction, and to clarify the design of our text experiments at the beginning of section 4.

---

## Referee Comment (RC2) · Anonymous Referee #3 · 7 May 2019

General comments: This manuscript presents a concept for the application of in situ electrostatic focusing to isothermally concentrate a continuously flowing aerosol stream of submicron particles, at ambient pressure. The authors demonstrated proof-of-concept, through theoretical calculation and laboratory measurement using a proto-type. This system may have potential implications in aerosol measurements under low particle concentration. I recommend publication of this manuscript with minor revison.

Specific comments

(1) Introduction: It would be better to include more studies that requiring size-selected by a DMA, which should be more related to the current study.

Results and Discussion:

(1) Line 3: "The observed enrichment is summarized in Figure... As shown in Figure", please indicate the specific figure number.

(2) From Figure 3, it can be seen that the enrichment factors are linearly related to the applied voltage for particles in the size range of 75 to 200 nm. I am afraid this size range does not cover the sub-micron particles in ambient.

(3) I wonder if other factors such as relative humidity and temperature in the system affect the enrichment.

(4) Given such a low enrichment factor observed in Figure 6, how could the authors extend the implications of this system.

---

## Author Comment (AC2) · 28 May 2019

**Reviewer 3 Response – Woo et al., "Concept for an electrostatic focusing device for continuous ambient pressure aerosol concentration."**

**General comments**

"This manuscript presents a concept for the application of in situ electrostatic focusing to isothermally concentrate a continuously flowing aerosol stream of submicron particles, at ambient pressure. The authors demonstrated proof-of-concept, through theoretical calculation and laboratory measurement using a prototype. This system may have potential implications in aerosol measurements under low particle concentration. I recommend publication of this manuscript with minor revison."

We thank the reviewer for their comments and feedback. Responses to specific comments are addressed below.

**Introduction**

"It would be better to include more studies that requiring size-selected by a DMA, which should be more related to the current study."

Several studies employ DMA-based selection to explore size-dependent properties on ambient (Mei et al., 2013; Thalman et al., 2017) or laboratory-generated (Ahern et al., 2016; Petters et al., 2006; Vaden et al., 2011) organic particles. In size-selecting a narrow cross-section of a polydisperse aerosol distribution, resulting aerosol concentration is subsequently low, and would benefit from in-situ concentration enrichment. Discussion of these applications will be added to the Introduction section.

**Results and Discussion**

"Line 3: "The observed enrichment is summarized in Figure... As shown in Figure", please indicate the specific figure number."

These lines refer to Figure 6; we will update the text to reflect this.

"From Figure 3, it can be seen that the enrichment factors are linearly related to the applied voltage for particles in the size range of 75 to 200 nm. I am afraid this size range does not cover the sub-micron particles in ambient."

The range of tested aerosol diameters is consistent with a number of laboratory-based chamber experiments that utilize monodisperse organic aerosol. (Frosch et al., 2011; Vaden et al., 2011) Field campaigns in polluted urban environments, have also observed ambient mean particle diameter within the upper range of our tested diameters. (Klejnowski et al., 2013; Xu et al., 2016) As such, we believe that the sizes considered in this study are both relevant and appropriate for a wide range of applications.

While smaller particle diameters were not experimentally assessed, we estimate values of applied voltage that will satisfy the  $\kappa \ge 1$  conditions necessary to achieve concentration for smallermobility aerosol; the results of these calculations for 30nm particles are demonstrated in Figure 2. For larger, coarse-mode aerosol, inertial forces are likely to dominate beyond the point where electrostatic deflection will achieve any meaningful concentration enhancement effects (i.e. $\kappa <$  1.) However, our proposed inlet geometry applies an electric field to a virtual impactor geometry; as a result, under such conditions where inertial forces govern aerosol motion, this system would hypothetically still be able to achieve concentration enhancement.

**"I wonder if other factors such as relative humidity and temperature in the system affect the enrichment."**

Under the temperature ranges where the inlet system is generally expected to be implemented (i.e. standard or near-standard temperatures and pressures,) relative humidity and temperature are not expected to significantly affect the electric field resulting from the electrical potential differences in the inlet system, or aerosol charging behavior. However, while humidity is not expected to significantly affect the charging properties of low-hygroscopicity aerosols, such as those described in this work, it does play a factor in the corona discharge behavior of the unipolar charger used to generate the positively charged aerosol to be deflected; at higher RH values (>80%,) corona onset voltages decrease and discharge currents increase, implying lower efficiencies in charging aerosol.(Yawootti et al., 2015) As a result, the maximum potential enrichment may be reduced, though the exact dependences on RH are outside the scope of this work.

**"Given such a low enrichment factor observed in Figure 6, how could the authors extend the implications of this system[?]"**

As the reviewer correctly notes, the inlet system is presented as a proof-of-concept rather than as a standalone technology. However, the observed concentration enhancement demonstrated by our prototype understate the theoretical extents of enrichment possible using electrostatic deflection (Figure 3); our estimations for concentration enhancement reach as high as 65% within our explored ranges of aerosol sizes and applied voltages. As seen in Figure 6, the observed reduction in enrichment did not uniformly apply across different aerosol diameters, leading to opposite trends at different applied voltages. This nonuniformity implies that several competing factors are contributing to these losses.

As discussed in our Results and Discussion, we attribute a significant amount of our losses to electric field distortions within the inlet system. Beyond the assumptions already described in the manuscript regarding interactions between the immersion lens and other solid surfaces in the inlet system, the immersion lens geometry assumes negligible edge field effects and that the two cylindrical tubes comprising the lens are identically coaxial. In practice, the implementation of the two tubes may have introduced translational and angular asymmetries not represented in the idealized immersion lens geometry, leading to spherical field aberration effects (e.g. coma, astigmatism). (Heddle, 2000) The resulting radial and angular field asymmetry may cause deflected particles to impact the inside of the collection probe, rather than into the minor flow. Furthermore, any physical irregularity in the surface of the lens tubing (i.e. machining imperfections, surface roughness, etc.) would lead to further reduced focusing effects as their resultant electric field deviates from idealized circumstances. As enrichment was still measurable in the inlet system without accounting for these electric field nonidealities, we believe that the inherent premise of using electrostatic deflection to concentrate aerosol streams is viable, especially as future iterations improve upon these distortion effects.

Discussion of the aforementioned additional sources of charge field distortion will be added to the Results and Discussion section of the manuscript.

**References**

Ahern, A. T., Subramanian, R., Saliba, G., Lipsky, E. M., Donahue, N. M. and Sullivan, R. C.: Effect of secondary organic aerosol coating thickness on the real-time detection and characterization of biomassburning soot by two particle mass spectrometers, Atmos. Meas. Tech, 9, 6117–6137, doi:10.5194/amt-96117-2016, 2016.

Frosch, M., Bilde, M., DeCarlo, P. F., Jurányi, Z., Tritscher, T., Dommen, J., Donahue, N. M., Gysel, M., Weingartner, E. and Baltensperger, U.: Relating cloud condensation nuclei activity and oxidation level of  $\alpha$  -pinene secondary organic aerosols, J. Geophys. Res. Atmos., 116(D22), D22212, doi:10.1029/2011JD016401, 2011.

Heddle, D. W. O.: Electrostatic Lens Systems. Institute of Physics Publishing, London, 2000.

Klejnowski, K., Krasa, A., Rogula-Kozłowska, W. and Błaszczak, B.: Number Size Distribution of Ambient Particles in a Typical Urban Site: The First Polish Assessment Based on Long-Term (9 Months) Measurements, Sci. World J., 2013, 1–13, doi:10.1155/2013/539568, 2013.

Mei, F., Setyan, A., Zhang, Q. and Wang, J.: CCN activity of organic aerosols observed downwind of urban emissions during CARES, Atmos. Chem. Phys. Discuss., 13(4), 9355–9399, doi:10.5194/acpd-13-9355-2013, 2013.

Petters, M. D., Kreidenweis, S. M., Snider, J. R., Koehler, K. A., Wang, Q., Prenni, A. J. and Demott, P. J.: Cloud droplet activation of polymerized organic aerosol, Tellus B Chem. Phys. Meteorol., 58(3), 196–205, doi:10.1111/j.1600-0889.2006.00181.x, 2006.

Thalman, R., De Sá, S. S., Palm, B. B., Barbosa, H. M. J., Pöhlker, M. L., Alexander, M. L., Brito, J., Carbone, S., Castillo, P., Day, D. A., Kuang, C., Manzi, A., Ng, N. L., Iii, A. J. S., Souza, R., Springston, S., Watson, T., Pöhlker, C., Pöschl, U., Andreae, M. O., Artaxo, P., Jimenez, J. L., Martin, S. T. and Wang, J.: CCN activity and organic hygroscopicity of aerosols downwind of an urban region in central Amazonia: seasonal and diel variations and impact of anthropogenic emissions, Atmos. Chem. Phys, 17, 11779–11801, doi:10.5194/acp-17-11779-2017, 2017.

Vaden, T. D., Imre, D., Beránek, J., Shrivastava, M. and Zelenyuk, A.: Evaporation kinetics and phase of laboratory and ambient secondary organic aerosol., Proc. Natl. Acad. Sci. U. S. A., 108(6), 2190–5, doi:10.1073/pnas.1013391108, 2011.

Xu, X., Zhao, W., Zhang, Q., Wang, S., Fang, B., Chen, W., Venables, D. S., Wang, X., Pu, W., Wang, X., Gao, X. and Zhang, W.: Optical properties of atmospheric fine particles near Beijing during the HOPE-J3A campaign, Atmos. Chem. Phys., 16(10), 6421–6439, doi:10.5194/acp-16-6421-2016, 2016.

Yawootti, A., Intra, P., Tippayawong, N. and Rattanadecho, P.: An experimental study of relative humidity and air flow effects on positive and negative corona discharges in a corona-needle charger, J. Electrostat., 77, 116–122, doi:10.1016/J.ELSTAT.2015.07.011, 2015.

---

## Author Response (AR2)

**Manuscript Edits – Woo et al., "Concept for an electrostatic focusing device for continuous ambient pressure aerosol concentration."**

*5 June 2019.*

We thank the editor for his feedback and input. The following changes have been made to the manuscript, as per his suggestions. All edits have been highlighted in the included annotated, updated manuscript.

- Text in the Introduction and Prototype Design sections have been adjusted and reworded for improved clarity.
- Figure 6 has been altered such that each applied voltage dataset for enrichment vs. particle diameter is a different color, for improved readability.
- Minor typographical corrections have been made.